# Informal Employment and Poor Mental Health in a Sample of 180,260 Workers from 13 Iberoamerican Countries

**DOI:** 10.3390/ijerph19137883

**Published:** 2022-06-27

**Authors:** Michael Silva-Peñaherrera, Paula Santiá, Fernando G. Benavides

**Affiliations:** 1Observatorio Iberoamericano de Seguridad y Salud en el Trabajo, Organización Iberoamericana de Seguridad Social (OISS), C. de Velázquez, 105, 28006 Madrid, Spain; michael.silva@upf.edu (M.S.-P.); psantia@psmar.cat (P.S.); 2Center for Research in Occupational Health, IMIM (Hospital del Mar Medical Research Institute), Universitat Pompeu Fabra, Doctor Aiguader, 88, 08003 Barcelona, Spain; 3CIBER Epidemiología y Salud Pública (CIBERESP), Av. Monforte de Lemos, 3-5, 28029 Madrid, Spain; 4Preventive Medicine and Public Health Training Unit PSMar-UPF-ASPB (Parc de Salut Mar—Pompeu Fabra University—Agència de Salut Pública de Barcelona), Passeig Maritim 25, 08003 Barcelona, Spain

**Keywords:** informal employment, mental health, health inequalities, occupational health, working conditions surveys

## Abstract

The aim of this study is to estimate the association between employment conditions and mental health status in the working population of Iberoamerica. In this cross-sectional study, we pooled individual-level data from nationally representative surveys across 13 countries. A sample of 180,260 workers was analyzed. Informality was assessed by social security, health affiliation, or contract holding. Mental health was assessed using several instruments. We used Poisson regression models to estimate the contribution of informality to poor mental health by sex and country, adjusted by sociodemographic and work-related characteristics. Then, we performed a meta-analysis pooling of aggregate data using a random-effects inverse-variance model. Workers in informal employments showed a higher adjusted prevalence ratio (aPR) of poor mental health than those in formal employment in Peru (aPR men 1.5 [95% confidence intervals 1.16; 1.93]), Spain (aPR men 2.2 [1.01; 4.78]) and Mexico (aPR men 1.24 [1.04; 1.47]; women 1.39 [1.18; 1.64]). Overall estimates showed that workers in informal employment have a higher prevalence of poor mental health than formal workers, with it being 1.19 times higher (aPR 1.19 [1.02; 1.39]) among men, and 1.11 times higher prevalence among women (aPR 1.11 [1.00; 1.23]). Addressing informal employment could contribute to improving workers’ mental health.

## 1. Introduction

Paid work is a social determinant of health that can act to either promote or hinder wellness. It provides financial security, personal identity, and an opportunity to make a meaningful contribution to community life [1]. However, there are stressful characteristics of work related to demands, control, and support that can act to the detriment of workers’ health [2].

Informal employment is defined as a non-regulated placement in the labor market that involves an undocumented arrangement between employee and employer in which there is no labor regulation, income taxation, social protection or entitlement to certain benefits, such as advance notice of dismissal, or unemployment and sick leave benefits [3]. Workers in informal employment may be exposed, to a greater extent, to the stressful characteristics of work and this may be reflected in their mental health status.

Underemployment [4,5], an unfavorable psychosocial environment with high demands and low control or an effort-reward imbalance [6,7,8], low procedural or relational justice [9,10,11], and non-permanent work have been associated with mental disorders [12,13,14,15,16,17]. However, evidence of the impact of informality on mental health is limited [18] and most previous studies were conducted in upper income countries [16,19,20,21].

The Iberoamerican community is composed of 22 Spanish- and Portuguese-speaking countries (19 Latin American countries, plus Spain, Portugal and Andorra). The relations between Iberoamerican countries have been intensifying, especially multilateral agreement on social security, which has been led by the Ibero-American Social Security Organisation (OISS by its acronym in Spanish) [22]. The current and foreseen trade agreements between the European Union and Latin America require the guarantee of conditions of equality and basic elements for sustainability, including policies that ensure decent and productive work [23]. Evidence from Iberoamerica is scarce and controversial. Previous studies showed that Brazilian workers in informal employment (categorized as self-employed and underemployed) have a higher prevalence of common mental disorders than those in formal employment, and that this association is stronger among women [16,24]. In Central America, workers lacking social security coverage showed a higher prevalence of poor mental health [25]. On the contrary, results from Colombia show that most informal workers perceive their health as normal, good, or very good and are satisfied with their quality of life. However, these results have not been compared to those of workers in formal employment [26]. Results from Chile show that the prevalence of poor mental health is higher among informal workers when compared to formal workers only in the case of male dependent workers, and that there are no differences among women [18].

The aim of this study is to assess the relationship between poor mental health and informal employment among a sample of workers from 13 Iberoamerican countries.

## 2. Materials and Methods

### 2.1. Study Design

Cross-sectional.

### 2.2. Data Source and Study Participants

Nationally representative surveys of household, health, or working conditions were identified for 13 Iberoamerican countries (Table 1). These 13 countries were selected for the study because data could be found that met the following criteria. Eligible surveys were performed after 2012 and included individual-level data on employment conditions and mental health assessed by a valid instrument. When more than one survey for a country complied with inclusion criteria, the most recent was chosen.

According to methodological reports from these surveys, all interviews were performed face-to-face at the interviewees’ houses. Our sample was restricted to those who had worked during the week preceding the interview, or were absent from work whether due to leave, illness, strike or attendance at a work-related course. This resulted in a sample of 180,260 workers aged 15 and over.

### 2.3. Variables

#### 2.3.1. Outcome

Mental health status was assessed by five different instruments. For the World Health Organization—Five Well-Being Index (WHO-5), poor mental health was defined as a score under 13 [27]; for the 9-item Patient Health Questionnaire (PHQ-9), as a score of 10 or more [28]; for the psychological domain of the Abbreviated World Health Organization Quality of Life questionnaire (WHOQOL-Bref), as a score under 60 [29]; for the 12-item General Health Questionnaire (GHQ-12) following the GHQ usual scoring method (0–0–1–1), as a score of 3 or more [30]; and for the affect domain of the Washington Group Extended Set on Functioning (WG-ES), as a score of 4 in either depression or anxiety scales [31].

#### 2.3.2. Work-Related Variables

Regarding employment conditions, in order to operationalize the variable of informal employment, according to the ILO approach [32], social security or health affiliation, or having a contract, were the criteria used based on the information available. Workers were classified according to their social security affiliation as either formal (affiliated) or informal (not affiliated). This was the case for Argentina, countries in Central America, and Peru. When information on social security was not available, classification was based on affiliation with a health system: workers who were entitled to health services as holders were considered formal and those not entitled or entitled as beneficiaries for other family members were considered informal. This was the case for Brazil. In the case of Chile, Mexico, Portugal, and Spain, due to data availability, employment status was based on contract holding. Workers holding a contract were considered formal and those working without a contract were considered informal (Appendix A).

Regarding labor relationship, workers were classified as an employee, self-employed or an employer. Occupational category was first transformed from local classification into the International Standard Classification of Occupation (ISCO-08) [33] and then grouped into four categories: non-manual skilled (managers, professionals, technicians and associate professionals), non-manual non-skilled (clerical support and services and sales workers), manual skilled (skilled agricultural, forestry and fishery workers, craft and related trades workers, and plant and machine operators and assemblers), and manual non-skilled (elementary occupations). Further aggrupation into manual and non-manual was performed. Workers in the armed forces were excluded from the analysis.

#### 2.3.3. Socio-Demographic Variables

Age comprised four categories (younger than 24; 25–44; 45–64; and over 65 years old). Highest educational level achieved was categorized into less than primary school; primary school; middle school; and higher education. Marital status comprised two categories: married or cohabiting; and single, separated, or widowed.

### 2.4. Analysis

At the country level, we describe the population distribution by frequencies and weighted proportions (*N*%). We estimated the prevalence of poor mental health with a 95% confidence interval (95% CI) by work-related and socio-demographic characteristics, and estimated the association between employment condition and poor mental health by crude prevalence ratios (cPR) using Poisson regression with robust variance. Models were adjusted by age, marital status, education, occupational category, and labor relationship in the case of Argentina, Chile, Peru, Brazil and countries in Central America. Models for Spain and Portugal were adjusted by age, marital status, education, and occupational category, and models for Mexico were adjusted by age, marital status, and education due to data availability. Country-specific survey weights were applied. All analyses were stratified by sex.

Second, a meta-analysis [34] pooling of aggregate data from each country using a random-effects inverse-variance model was performed to assess the overall effect of employment condition on mental health status in the region. The analysis was stratified by a mental health measurement tool.

## 3. Results

### 3.1. Population Characteristics

Workers in informal employment represent more than 40% (*N* = 83,781) of the working population in Iberoamerica according to the surveys included in this study. Informality was highest in Guatemala (men: 88.0% [85.1; 90.4]; women: 88.3% [85.0; 91.0]) and lowest in Spain (men: 3.9% [2.70; 5.10]; women: 6.8% [5.27; 8.25]). Spain also showed the lowest proportion of workers in manual occupations (men: 43.5% [40.6; 46.3]; women: 23.6% [21.2; 25.9]), while Honduras showed the highest (men: 82.5% [79.6; 85.0]; women: 57.3 [52.6; 61.9]). Men showed higher rates of working in manual occupations than women across all countries. Approximately 17% of workers (*N* = 31,135) did not finish primary school, this figure being below 3% in Spain and Portugal and above 25% for Brazil. Women represented a lower proportion of unschooled workers than men, except for in the cases of Peru and Guatemala (Appendix A).

### 3.2. Mental Health Distribution

Among the countries using GHQ-12, the lowest prevalence of poor mental health was found in Guatemalan men (8.1% [5.94; 11.0]) and Salvadoran women (11.0% [8.5; 13.4]), while the highest was found in Peruvian workers (men 27.9% [25.5; 30.4); women 32.5% [29.9; 35.3]). Among the countries using WHO-5, Argentine workers showed a considerably higher prevalence of poor mental health (men: 15.6% [14.4; 16.8]; women: 21.9 [20.2; 23.5]) than workers in Spain (men 9.9% [8.4; 11.8]; women 13.7% [11.8; 15.9]) and Portugal (men: 11.6% [8.44; 15.8]; women 14.1% [11.1; 17.7]). Women consistently showed a higher prevalence of poor mental health than men across countries, except for Costa Rica (men 15.1% [10.7; 19.6]; women 14.8 [11.0; 18.7]).

The differences between the sexes were greater for Brazil, with a prevalence ratio of women: men (W:M) of 2.89 and an absolute difference of 8.70 percentage points (pp) (men 4.6% [4.2; 5.0]; women 13.3% [12.6; 14.1]), followed by Mexico and Guatemala, regarding relative difference (W:M Mexico 1.76 Guatemala 1.65), and by Argentina and Chile, regarding absolute differences (Argentina 6.30 pp, Chile 6.0 pp).

The least educated workers showed a higher prevalence of poor mental health than the most educated across regions, except for Panamanian and Brazilian male workers. Differences in the prevalence of poor mental health between the lowest and highest educational categories were greater for women than for men, except in the cases of El Salvador (men 4.8 pp, women 2.3 pp) and Spain (men 10.2 pp, women 7.2 pp) (Table 2 and Appendix A).

### 3.3. Informality and Poor Mental Health

Informal workers showed a higher prevalence of poor mental health than their counterparts across most countries and both sexes. However, formal male workers from Panama (27.0% vs. 19.2%), Portugal (11.1% vs. 10.7%), and Brazil (4.8% vs. 4.5%) and formal female workers from El Salvador (14.5% vs. 10.5%) showed a higher prevalence of poor mental health than informal workers (Table 2 and Appendix A).

The prevalence ratios of poor mental health between informal and formal workers were significant in the case of male workers from Costa Rica (cPR 1.80 [1.00; 3.25]), Peru (cPR 1.63 [1.31; 2.01]), and Mexico (cPR 1.17 [1.00; 1.37]) and female workers from Peru (cPR 1.45 [1.15; 1.83]), Chile (cPR 1.51 [1.10; 2.06]), Brazil (cPR 1.17 [1.02; 1.34]), and Mexico (cPR 1.55 [1.34; 1.79]). Overall crude estimates show that men and women in informal employment had a 22% and 27% higher prevalence of poor mental health than their formal counterparts, respectively (cPR men 1.22 [1.03; 1.39]; women 1.27 [1.12; 1.43]) (Table 3).

In adjusted models, men in informal employment showed a higher prevalence of poor mental health than formal workers in Peru (aPR 1.5 [1.16; 1.93]), Spain (aPR 2.2 [1.01; 4.78]), and Mexico (aPR 1.24 [1.04; 1.47]). In the case of women, the results are statistically significant only in the case of Mexico (aPR 1.39 [1.18; 1.64]). Overall adjusted estimates continue to show a higher prevalence or poor mental health in informal workers when compared with formal workers, although these values are lower than the crude estimates. Overall, among men, informal workers showed a 1.19 times higher prevalence of poor mental health than formal workers (aPR 1.19 [1.02; 1.39]). Among women, informal workers showed a 1.11 higher prevalence than formal workers (aPR 1.11 [1.00; 1.23]) (Table 3 and Figure 1).

## 4. Discussion

Our results show that, overall, Iberoamerican workers in informal employment have a higher prevalence of poor mental health than workers in formal employment. However, the results are heterogeneous among countries and, in most cases, we could not establish a robust association between informality and poor mental health. These results may be due to various reasons: The inaccurate measurement of mental health status; the inaccurate measurement of informality; unobserved confounding factors concealing a real association; or a non-existing association between informality and poor mental health. It is important to note that informal employment is related not only to poor working and employment conditions, but also to poor living conditions, social and economic vulnerability, and poor environmental conditions. All of these factors could affect the association. Previous studies showed that national indicators of GDP, CO_2_ emissions, educational attainment, life expectancy, rates of poverty, and women’s labor market participation are strongly associated with the rate of informality [35].

### 4.1. Mental Health Status Measurement

Regarding the prevalence of poor mental health, the information found was heterogeneous, with poorly standardized records, making it difficult to establish accurate regional comparisons. The mental health questionnaires used in each survey measure different concepts of health. Take, for example, PHQ-9 and WHO-5. The former scores each of the nine DSM-IV criteria for depression with questions such as “Have you felt depressed or hopeless?”, while the latter assesses psychological well-being with statements such as “I have felt cheerful and in good spirit”. However, other studies have established measurement similarities between PHQ-9 and GHQ-12 [36,37], WHO-5 and GHQ-12 [38], WHO-5 and PHQ-9 [39], and PHQ-9 and the depression domain of WG-ES [40]. Besides differences across questionnaires, differences in social, cultural and historical background across countries may influence workers’ perception of their mental health and quality of life, both being closely related to one’s expectations. Thus, even when using the same questionnaire, mental health evaluation requires a country-specific analysis, and questionnaire validation in each country is encouraged [41].

In addition, estimates of poor mental health may be underestimated for two reasons: first, the prevalence of somatic symptom in Latin America is high, while mood symptoms are underrepresented [42]. Hence, we would expect low rates of self-identified depression. Second, GHQ-12, although used extensively, may not be the best screening tool [43].

The prevalence of poor mental health in Mexico is remarkably low; however, the case definition refers to mental conditions causing high disability. Other studies have already reported similar rates when severity is considered [44,45].

### 4.2. Exposure Measurement

Informality was defined according to ILO specifications, and rates of informal employment described hereby are similar to those depicted by ILO [46], implying general measurement accuracy, except for El Salvador and Brazil where we found higher rates, and Spain and Chile where we found lower rates. Nevertheless, other authors have already argued that this definition may be insufficient to analyze the context of the Latin American working population [47]. Furthermore, employment conditions on the main job are used to depict informality, but a person can simultaneously have two or more formal and/or informal jobs. The proportion of workers holding multiple jobs varied in the sample, from 3% in Mexico to around 19% in Argentinian women and Peruvian men, and was not available for Chile. The characteristics of these other jobs could not be ascertained.

Formal workers may not be as protected as we would expect, diminishing the differences between formal and informal workers. Workers at formal jobs may also be exposed to other work-related risk factors for poor mental health, such as an increased number of working hours, unpaid overtime, poor job security, poor satisfaction with one’s work culture and a feeling of a lack of support, low income, shift work, and night work [48,49]. Moreover, the provision of mental health services represents less than 1% of government health spending in low- and middle-income countries, and rates of availability and uptake for mental health services remain very low [50]. Therefore, health and social services may not meet the mental health needs of formal workers in the region. On the other hand, workers may have other support systems outside social security and state resources.

### 4.3. Unobserved Confounders

Other life circumstances may have a greater impact on mental health than employment condition and may act as unobserved confounders. Exposure to poverty, food insecurity, trauma, housing instability, and decreased social support have all been proven to affect one’s physical and mental well-being [51,52]. Race and migrant status have also been proven to highly affect mental health. Information on these circumstances could not be ascertained for each country and was, therefore, not included in the analysis. Future studies should explore the effect of these living conditions on the association between informality and poor mental health.

### 4.4. Education

When adjusting for the highest education level achieved, the magnitude of the association between informality and poor mental health decreased, implying that part of that association was explained by education. Low education may be associated with poor mental health in two ways: early onset disorders have been shown to impact school dropout, and school dropout has been proposed as a cause of internalizing disorders [53,54,55].

### 4.5. Gender

The association between informality and poor mental health was higher and stronger for men than for women when adjusting for sociodemographic and other work-related variables. The unequal involvement of women in unpaid labor can wield deleterious effects on their mental health and may explain some of the differences found between the sexes [56]. Information on care load and domestic work was only available for questionnaires from Brazil and Chile. In Brazil, women were in charge of caring for dependents (children, the elderly, the sick or people with special needs) more frequently than men, and those in charge of care showed a higher prevalence of poor mental health. In Chile, the results were similar, with a larger percentage of women in charge of caring for dependents and domestic work than men. Women with double presence showed a significantly higher prevalence of poor mental health.

### 4.6. Strengths and Limitations

The findings should be considered in light of the limitations of this study. First, in all cases, survey participants were limited to those with permanent residence and sufficient proficiency in the official language (Spanish or Portuguese), and therefore the prevalence of poor mental health may be underestimated by the exclusion of the most vulnerable workers, such as immigrants. Second, the surveys included relied solely on retrospective self-reports; hence, the data are subject to recall bias and the willingness to disclose information truthfully. Given the generally stigmatizing nature of mental health disorders, it is likely that reporting bias leads to an underestimation of prevalence. The cross-sectional design of the study prevented us from making inferences regarding causality or directionality between socio-demographic and work-related correlates and mental health status. To operationalize informal employment, and according to the ILO approach [32], different criteria were used for each country, depending on the information available: affiliation with social security, with a health system or with having a contract. However, in many countries, workers with a contract must be affiliated with a pension or health system, and thus not being affiliated with social security or a health system is tantamount to the same concept: informality. Another important limitation is that mental health status was assessed using five different instruments. However, all instruments used were validated, widely recognized and reliable measures of mental health. In addition, we measured the prevalence ratio of poor mental health between formal and informal workers within countries using the same instrument. Finally, given that the data come from different countries and therefore from different sources, the methodologies might have differed slightly. In addition, the surveys were conducted in different years, between 2012 and 2018. However, this study used data from each country to measure the difference between the mental health of informal and formal workers within countries. All these limitations encourage caution when interpreting the results.

Despite these limitations, to our knowledge this is the first study to report on the mental health status of the working population of Iberoamerica as a whole and to highlight the deleterious effect of informality on workers’ mental health, regardless of other sociodemographic and work-related characteristics. In this study, we provide evidence, based on the most recent data available, for local occupational health planning and a starting point for further surveillance in the region.

## 5. Conclusions

Addressing informal employment could contribute to improving the mental health of workers in Iberoamerica. Work is a key determinant of workers’ mental health, but other determinants such social and living conditions should be included in prevention policies. Given the great heterogeneity among surveys in the region, we recommend further research to incorporate common indicators of mental health and informality in order to enable comparisons. Additionally, variables such as hours dedicated to unpaid work at home and care for dependents should be included to better analyze the interaction between work, gender, and mental health.

## Figures and Tables

**Figure 1 ijerph-19-07883-f001:**
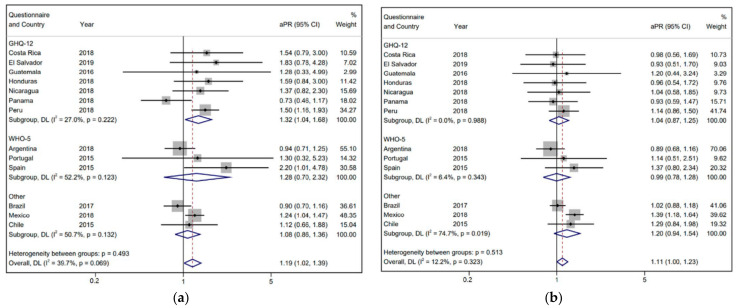
Forest plots showing adjusted prevalence ratios (aPR) of poor mental health between informal and formal workers of Iberoamerica, by country and mental health questionnaire, and overall estimates from the meta-analysis. (**a**) Men. (**b**) Women. The dashed red line shows the overall result.

**Table 1 ijerph-19-07883-t001:** Data source and mental health measurement tool.

Questionnaire	Country	Survey	Year	*N*
GHQ-12 ^1^	Costa Rica	II Central American Survey of Working and Health Conditions	2018	1503
	El Salvador	II Central American Survey of Working and Health Conditions	2018	1374
	Guatemala	II Central American Survey of Working and Health Conditions	2018	1510
	Honduras	II Central American Survey of Working and Health Conditions	2018	1507
	Nicaragua	II Central American Survey of Working and Health Conditions	2018	1500
	Panama	II Central American Survey of Working and Health Conditions	2018	1505
	Peru	National Survey on Working Conditions, Safety and Health	2015	3113
WHO-5 ^2^	Argentina	National Survey of Workers on Employment Conditions, Health and Safety	2018	8966
	Portugal	European Working Conditions Survey, 6th edition	2015	896
	Spain	European Working Conditions Survey, 6th edition	2015	3291
WHOQOL-Bref ^3^	Chile	National Quality of Life and Health Survey	2016	3126
PHQ-9 ^4^	Brazil	National Health Survey	2019	52,282
WG-ES ^5^	Mexico	National Household Survey	2017	99,687
	**Total**			**180,260**

^1^ GHQ-12: 12-item General Health Questionnaire. ^2^ WHO-5: World Health Organization—Five Well-Being Index. ^3^ WHOQOL-Bref: Abbreviated World Health Organization Quality of Life Questionnaire. ^4^ PHQ-9: Patient Health Questionnaire. ^5^ WG-ES: The Washington Group Extended Set on Functioning.

**Table 2 ijerph-19-07883-t002:** Prevalence of poor mental health (%) according to work and sociodemographic variables, by questionnaire, country, and sex.

	Questionnaire and Country												
	GHQ-12							WHO-5			Other		
Men	CR	ES	GU	HO	NI	PA	PE	AR	PO	SP	CH	BR	ME
**Total**	15.1	8.8	8.1	19.4	21	22.8	27.9	15.6	11.6	9.9	15.8	4.6	3.7
**Employment condition ^1^**													
Formal	11.6	5.9	5.9	11.7	14.7	27.0	20.5	15.5	11.1	9.8	15.1	4.8	3.0
Informal	20.9	9.4	8.2	20.6	22.9	19.2	32.3	15.7	10.7	19.7	17.1	4.5	3.5
**Labor relationship**													
Employee	11.1	9.5	7.2	18.2	18.0	25.6	25.9	14.7	11.2	10.2	16.6	4.3	3.2
Self-employed or employer	19.6	8.5	8.5	20.4	23.2	20.7	30.8	17.6	12.7	9.3	13	4.9	5.2
**Occupational category ^2^**													
Non-manual	14.6	10.1	3.7	12.7	10.9	21.5	23.3	15.6	9.1	9.9	12.3	5.5	-
Manual	15.3	8.6	9.8	20.8	23.5	23.1	32.3	15.8	14.1	10.0	17.8	4.0	-
**Age, years**													
Under 24	15.6	8.9	4.5	13.8	12.6	20.6	23.6	13.6	4.6	4.9	7.5	4.4	1.4
25–44	14.2	9.5	13.2	15.0	21.5	22.7	23.6	14.9	11.8	10.2	14.8	4.4	3.4
45–64	19.4	9.3	5.2	26.7	27.6	24.0	32.6	18.1	12.8	10.4	17.8	5.0	5.3
≥65	2.2	7.0	7.8	30.8	19.2	22.8	46.5	8.2	-	-	24.9	3.3	6.2
**Education**													
Less than primary school	8.1	13.3	13.3	32.9	37.1	19.4	45.7	18.6	15.6	20.3	28.5	4.5	5.0
Primary school	18.9	10.1	8.6	20.8	18.3	21.1	41.3	16.2	14.7	8.9	16.7	4.6	3.9
Secondary school	12.9	6.6	7.1	15.0	21.0	23.9	26.5	13.5	13.3	9.0	15.7	4.6	3.3
Higher education	7.7	8.5	2.9	4.5	15.2	22.4	21.5	16.7	3.4	10.1	10.8	4.7	4.1
**Marital status**													
Married or co-habiting	15.3	5.9	8.0	21.1	21.7	22.3	29.0	15.6	12.2	10.6	14.6	4.1	4.2
Single. divorced or widowed	14.7	14.0	8.3	15.7	19.2	23.4	26.6	15.6	11.4	5.8	17.4	5.9	2.6
**Women**	**CR**	**ES**	**GU**	**HO**	**NI**	**PA**	**PE**	**AR**	**PO**	**SP**	**CH**	**BR**	**ME**
**Total**	14.8	11.0	13.4	23.2	21.4	25.3	32.5	21.9	14.1	13.7	21.8	13.3	6.5
**Employment condition ^1^**													
Formal	14.3	14.5	7.3	18.4	19,0	24.1	24.3	21.4	12.9	12.4	18.9	11.9	4.6
Informal	15.4	10.5	14.2	23.9	22.3	26.8	35.3	22.4	21.9	18.3	28.5	13.9	7.1
**Labor relationship**													
Employee	12.5	13.9	9.7	21.6	21.7	24.0	27.3	20.6	13.5	12.9	20.3	12.9	5.5
Self-employed or employer	17.0	9.9	15.2	24.4	21.1	27.8	39.8	25.7	17.1	18.9	28.5	14.4	9.1
**Occupational category ^2^**													
Non-manual	16.2	13.0	10.5	25.2	17.4	23.2	28.9	20.7	11.5	13.5	19.8	13.1	-
Manual	13.5	9.3	17.4	21.7	25.5	28.8	40.8	25.3	19.5	14.5	27.9	13.7	-
**Age. years**													
under 24	7.3	11.6	9.5	19.9	21.5	19.7	25.5	24.3	15.7	2.7	16.4	15.5	2.3
25–44	17.1	12.5	15.9	22.6	20.8	26.1	30.6	22.7	7.9	12.4	17.2	13.1	5.5
45–64	14.1	8.7	16.7	24.8	22.4	24.3	37.1	21.0	19.0	16.8	30.9	13.8	9.9
≥65	11.5	11.7	16.5	45.2	23.3	53.7	55.6	13.1	30.5	15.0	21.1	8.9	11.8
**Education**													
Less than primary school	56.0	16.5	26.8	48.2	37.7	36.5	51.1	26.2	47.7	20.2	38.9	15.5	10.8
Primary school	11.3	8.5	13.5	27.6	24.1	35.7	41.3	21.8	20.1	27.4	27.3	15.5	8.3
Secondary school	17.2	11.4	11.6	17.2	19.5	25.3	32.6	24.8	14.1	11.7	19.9	12.3	5.4
Higher education	11.8	14.2	5.0	9.3	13.9	20.0	22.8	19.5	7.1	13.0	16.9	11.5	5.2
**Marital status**													
Married or co-habiting	13.8	10.3	14.8	23.4	20.9	25.1	30.7	21.3	13.1	13.2	20.4	12.6	6.2
Single. divorced or widowed	16.0	12.0	11.7	23.0	21.9	25.7	34.5	22.8	19.4	14.5	22.3	14.4	6.9

GHQ-12: 12-item General Health Questionnaire. WHO-5: World Health Organization—Five Well-Being Index. CR: Costa Rica, ES: El Salvador, GU: Guatemala, HO: Honduras, NI: Nicaragua, PA: Panama, PE: Peru, AR: Argentina, PO: Portugal, SP: Spain, CH: Chile, BR: Brazil, ME: Mexico. Percentages are weighted by specific survey weights. ^1^ Employment condition in Spain, Portugal and Mexico was only available for employees. ^2^ Information on occupational category was not available for Mexico.

**Table 3 ijerph-19-07883-t003:** Prevalence ratio of poor mental health between formal (Ref.) and informal workers by country and sex.

Questionnaire	Country	Men		Women	
		cPR (95% CI)	aPR (95% CI)	cPR (95% CI)	aPR (95% CI)
GHQ-12	Costa Rica ^a^	**1.80 (1.00–3.25)**	1.54 (0.79–3.00)	1.08 (0.64–1.81)	0.98 (0.56–1.69)
	El Salvador ^a^	1.59 (0.73–3.48)	1.83 (0.78–4.28)	0.73 (0.45–1.17)	0.93 (0.51–1.70)
	Guatemala ^a^	1.38 (0.46–4.09)	1.28 (0.33–4.99)	1.95 (0.84–4.51)	1.20 (0.44–3.24)
	Honduras ^a^	1.76 (0.97–3.20)	1.59 (0.84–3.00)	1.30 (0.75–2.27)	0.96 (0.54–1.72)
	Nicaragua ^a^	1.56 (0.95–2.56)	1.37 (0.82–2.30)	1.17 (0.74–1.84)	1.04 (0.58–1.85)
	Panama ^a^	0.71 (0.50–1.02)	0.73 (0.46–1.17)	1.11 (0.75–1.65)	0.93 (0.59–1.47)
	Peru ^a^	**1.63 (1.31–2.01)**	**1.5 (1.16–1.93)**	**1.45 (1.15–1.83)**	1.14 (0.86–1.50)
WHO-5	Argentina ^a^	1.02 (0.83–1.24)	0.94 (0.71–1.25)	1.05 (0.87–1.26)	0.89 (0.68–1.16)
	Portugal ^b^	0.96 (0.28–3.31)	1.30 (0.32–5.23)	1.70 (0.80–3.63)	1.14 (0.51–2.51)
	Spain ^b^	2.01 (0.95–4.23)	**2.2 (1.01–4.78)**	1.48 (0.88–2.48)	1.37 (0.80–2.34)
WHOQOL-Bref	Chile ^a^	1.13 (0.77–1.65)	1.12 (0.66–1.88)	**1.51 (1.10–2.06)**	1.29 (0.84–1.98)
PHQ-9	Brazil ^a^	0.93 (0.76–1.15)	0.90 (0.70–1.16)	**1.17 (1.02–1.34)**	1.02 (0.88–1.18)
WG-ES	Mexico ^c^	**1.17 (1.00–1.37)**	**1.24 (1.04–1.47)**	**1.55 (1.34–1.79)**	**1.39 (1.18–1.64)**
Meta-analysis overall estimate		**1.22 (1.03–1.44)**	**1.19 (1.02–1.39)**	**1.27 (1.12–1.43)**	**1.11 (1.00–1.23)**

cPR: crude prevalence ratio. aPR: adjusted prevalence ratio. 95% CI: 95% confidence interval. ^a^ Model was adjusted for age, education, marital status, labor relationship and occupational category. ^b^ Model was adjusted for age, education, marital status and occupational category. ^c^ Model was adjusted for age, education and marital status. In bold are results statistically significant at the α 0.05 level.

## Data Availability

In this study, the datasets analyzed were publicly available from some countries, while others were obtained from third parties. Restrictions may apply to the availability of these latter datasets. Datasets are available at the following addresses: Argentina: https://www.argentina.gob.ar/srt/observatorio-srt/encuestas-salud-trabajo/ECETSS-2018 (accessed on 14 January 2022); Brazil: https://www.ibge.gov.br/estatisticas/sociais/saude/9160-pesquisa-nacional-de-saude.html?=&t=microdados (accessed on 14 January 2022); Chile: http://epi.minsal.cl/bases-de-datos/ (accessed on 14 January 2022); Centro America: The University of Texas Health Science Center at Houston (UTHealth) https://sph.uth.edu/research/centers/swcoeh/central-america/ (accessed on 14 January 2022) Available under request; Peru: Instituto Nacional de Salud: https://web.ins.gob.pe/ (accessed on 14 January 2022) Available under request; Mexico: https://www.inegi.org.mx/programas/enh/2017/#:~:text=La%20Encuesta%20Nacional%20de%20los.y%20las%20comunicaciones%20en%20los (accessed on 14 January 2022); Spain and Portugal: European Foundation for the Improvement of Living and Working Conditions. (2020). European Working Conditions Survey Integrated Data File. 1991–2015. [data collection]. 8th Edition. UK Data Service. SN: 7363. DOI: 10.5255/UKDA-SN-7363-8. Available under request.

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
