# Peer review of "Informal Employment and Poor Mental Health in a Sample of 180,260 Workers from 13 Iberoamerican Countries"

_ijerph, 2022, doi:10.3390/ijerph19137883_

Round 1

Reviewer 1 Report

This may be the first study to report on the mental health status of the working population of Iberoamerica as a whole and to highlight the deleterious effect of informality on worker´s mental health, regardless of other sociodemographic and work-related characteristics. This may be the primary contribution of the study. Thus, the following points and issues I concern become more important to strengthen the value of the study.

  1. Why was this area (i.e., Iberoamerica, if possible, I suggest a revision to Iberian America) selected in the study? If any special and common characteristics exist among these 13 countries besides the same language usage? As we know, Iberian America is a region in the Americas comprising countries or territories where Spanish or Portuguese are predominant languages (usually former territories of Portugal or Spain). Portugal and Spain are themselves included in some definitions, such as that of the Ibero American Summit and the Organization of Ibero American States. Nowadays, there are 23 countries and population in Europe and the Americas regarding Iberian America. In this study, however, only the data of 13 countries among these countries were analyzed. Was this because the availability of existing collected data? This should be clarified in Introduction or in Methods section.
  2. L28-29, when saying “Reducing informal employment should be a key objective if public health institutions wish to improve workers´ mental health.” The suggestion based on the results is relatively weak. The reasons for the demand for informal employment are quite complex. Although the mental health of informal workers was found to be poorer than that of formal workers, the authors should not intuitively suggest reducing informal employment, but how to deeply understand the current statuses of informal employment to improve the overall working conditions through effective interventions of policies and regulations.
  3. L50-62, At the end of the Introduction section, the authors cited some relevant studies and found that there were some contradictions among the studies. This is reasonable because these countries may have different living statuses and salary levels, and the survey and analysis methods between studies were also different. This may be the actual motivation for the study. However, as shown in Table 1, how can the authors be sure that there were no biases when using different survey methods for these countries? If this point is inherent, the study limitation should be noted at least.
  4. L24, Peru (aPR men 1.5 [1.16; 1.93]), Please spell out the full name when the term appears first time in the text. aPR, adjusted Prevalence Ratio; in addition, the 95% confidence intervals should also be noted for the usage of first time. However, in Abstract, the descriptions should be independent and complete. Similar suggestions are also available to the abbreviation cPR in the main text, not only explained in the footnote of the tables. These may be better mentioned in the subsection of Analysis (L122).
  5. Please follow the writing format of MDPI journal, such as section and subsection numbers, to enhance the readability of the paper.

Author Response

Reviewer 1

Comments and Suggestions for Authors: This may be the first study to report on the mental health status of the working population of Iberoamerica as a whole and to highlight the deleterious effect of informality on worker´s mental health, regardless of other sociodemographic and work-related characteristics. This may be the primary contribution of the study. Thus, the following points and issues I concern become more important to strengthen the value of the study.

Response: We greatly appreciate the reviewer’s positive comments and valuable suggestions. We have carefully revised the manuscript and have incorporated most of the reviewer’s suggestions.

COMMENT #1. Why was this area (i.e., Iberoamerica, if possible, I suggest a revision to Iberian America) selected in the study? If any special and common characteristics exist among these 13 countries besides the same language usage? As we know, Iberian America is a region in the Americas comprising countries or territories where Spanish or Portuguese are predominant languages (usually former territories of Portugal or Spain). Portugal and Spain are themselves included in some definitions, such as that of the Ibero American Summit and the Organization of Ibero American States. Nowadays, there are 23 countries and population in Europe and the Americas regarding Iberian America. In this study, however, only the data of 13 countries among these countries were analyzed. Was this because the availability of existing collected data? This should be clarified in Introduction or in Methods section.

Response The Ibero-American community is composed of 22 countries: 19 Spanish and Portuguesespeaking Latin American countries, and the Iberian Peninsula countries of Spain, Portugal and Andorra. These countries not only share cultural aspects, language and history, but are also integrated through political agreements that include economic and social aspects. In this sense, the European countries Spain, Portugal and Andorra are closely linked to the Latin American countries (1). These relations, which have been intensifying, are particularly successful for all parties in the area of social security and social protection. An area led by the Ibero-American Social Security Organisation. (OISS in spanish acronym : https://oiss.org/)(2). The current and foreseen trade treaties between the European Union and Latin America require in their clauses to guarantee decent and productive work, a basic element for the sustainability of any society.  

However, as we only included 13 countries, we have included the following paragraph in order to clarify this in the new version of the manuscript, (pag 2 line 50):

“The Ibero-American community is composed of 22 Spanish and Portuguese-speaking countries: 19 Latin American countries, plus Spain, Portugal and Andorra. The relations between Iberoamerican countries have been intensifying, especially multilateral agreement on social security that has been led by the Ibero-American Social Security Organisation. (OISS by its acronym in spanish). The current and foreseen trade agreements between the European Union and Latin America require guaranteeing conditions of equality and basic elements for sustainability, which implies policies that ensure decent and productive work.”

(pag 2 line 78)

“(…) These 13 countries were selected for the study because they have data according to the following criteria (…)”.

COMMENT #2.  L28-29, when saying “Reducing informal employment should be a key objective if public health institutions wish to improve workers´ mental health.” The suggestion based on the results is relatively weak. The reasons for the demand for informal employment are quite complex. Although the mental health of informal workers was found to be poorer than that of formal workers, the authors should not intuitively suggest reducing informal employment, but how to deeply understand the current statuses of informal employment to improve the overall working conditions through effective interventions of policies and regulations.

Response: Thank you for pointing that out. We agree, informal employment is quite complex. We

have therefore modified the sentence as follows (pag 1 line 28-29):

“Addressing informal employment could contribute to improving workers' mental health.”

COMMENT #3.  L50-62, At the end of the Introduction section, the authors cited some relevant studies and found that there were some contradictions among the studies. This is reasonable because these countries may have different living statuses and salary levels, and the survey and analysis methods between studies were also different. This may be the actual motivation for the study. However, as shown in Table 1, how can the authors be sure that there were no biases when using different survey methods for these countries? If this point is inherent, the study limitation should be noted at least.

Response: We agree that there may be methodological differences between surveys. However, in this study we do not directly compare results between countries but measure the difference between poor mental health of informal and formal workers within each country. However, to clarify this point, we have added a statement in the limitation section (pag 7 line 308)

“Another important limitation was that mental health status was assessed using five different instruments. However, all instruments used have been validated, widely recognized and reliable measures of mental health. In addition, we measured the prevalence ratio of poor mental health between formal and informal workers within countries using the same instrument. Finally, as the data come from different countries and therefore from different sources, the methodology might differ slightly. In addition, the surveys were conducted in different years, between 2012 and 2018. However, this study used data from each country to measure the difference between the mental health of informal and formal workers within countries. All these limitations should make us cautious when interpret-ing the results.”

COMMENT #4.  L24, Peru (aPR men 1.5 [1.16; 1.93]), Please spell out the full name when the term appears first time in the text. aPR, adjusted Prevalence Ratio; in addition, the 95% confidence intervals should also be noted for the usage of first time. However, in Abstract, the descriptions should be independent and complete. Similar suggestions are also available to the abbreviation cPR in the main text, not only explained in the footnote of the tables. These may be better mentioned in the subsection of Analysis (L122).

Response: We thank the reviewer for pointing out this. We have includedthe full name as follow.

(Pag 1 line 22)

“Workers in informal employments showed higher adjusted prevalence ratio (aPR) of poor mental health than those in formal employments in Peru (aPR men 1.5 [95% confidence intervals 1.16; 1.93]),”

(Pag 3 line 129)

“estimated the association between employment condition and poor mental health by crude prevalence ratios (cPR) using Poisson regression with robust variance”.

COMMENT #5.  Please follow the writing format of MDPI journal, such as section and subsection numbers, to enhance the readability of the paper.

Response: Thank you for the suggestion. We have used the MDPI journal template and used the section and subsection numbers.

Reviewer 2 Report

 This reviewer commends the authors for their study titled, “Informal employment and poor mental health in a sample of 180,260 workers from 13 Iberoamerican countries. However, the following comments and questions need to be addressed.

  1. Method  (page 3, lines 84-93)

Design

This study erroneously attempts to attribute population mental health to a single variable of informal employment.  According to the World Health Organization, “[c]ertain population subgroups are at higher risk of mental disorders because of greater exposure and vulnerability to unfavorable social, economic, and environmental circumstances, interrelated with gender.”1

Research Instruments

 “Mental health status was assessed by five different instruments. For the World Health Organisation- Five Well-Being Index (WHO-5), poor mental health was defined as a score under 13 [25]; for the 9-item Patient Health Questionnaire (PHQ-9) as a score of 10 or more [26]; for the psychological domain of the Abbreviated World Health Organisation  Quality of Life questionnaire (WHOQOL-Bref) as a score under 60 [27]; for the 12-item General Health Questionnaire (GHQ-12) following GHQ usual scoring method (0–0–1–1) as a score of 3 or more [28]; and for the affect domain of the Washington Group Extended 91 Set on Functioning (WG-ES) as a score of 4 in either depression or anxiety scales [29]. As stated in the preceding paragraph mental health status was assessed using five different instruments across the 13 Iberoamerican countries and yet attempts to convince readers that the different measurements can be used to represent the same outcome variable of mental health.

  1. Operational Definitions (page 3, lines 94-99)

Regarding employment conditions, workers were classified according to social security affiliation into formal (affiliated) or informal (not affiliated). This was the case for Ar gentina, countries in Central America and Peru. When information on social security was  not available, classification was based on affiliation to the health system: workers who are entitled to health services as holders were considered formal and those not entitled or entitled as beneficiaries to other family members were considered informal.” As stated in the preceding paragraph this study does not use the same operational definition for the term temporary employment across the 13 Iberoamerican countries in the study area, and yet attempts to convince readers that the different definitions represent the same independent variable.

  1. The Studies Does Not Account for Other Major Mental Health Variables   (page 3, lines 113 to 116

“Age was comprised into four categories (fewer than 24; 25-44; 45-64; and over 65 113 years old). Higher educational level achieved was categorized into less than primary 114 school; primary school; middle school; and higher education. Marital status was com- 115 prised into two categories: married or cohabiting; and single, separated or widowed.” As stated in the preceding paragraph this study does not provide insight into child labor and its impact on adult mental health (notice the first age bracket is fewer than 24, not fewer than 18 years) and also ignores the enormous impact of adverse childhood experiences on adult mental health. Thus the study erroneously assumes poor mental health is the result of getting informal employment as adults. But it is important to note that other variables not accounted for in this study such as childhood  adversity, discrimination (whether related to race/ethnicity, immigrant status, sexual orientation, and/or occupational status), nationality, and migration status have demonstrated significant negative impacts on mental health.2

  1. Ecological Fallacy

Studies using geographical units involving several countries are inherently characterized by analytical difficulties and can often suffer from an "ecological fallacy," which attributes collective characteristics to very dissimilar individuals or groups or the lack of agreement on the power and methods of data analysis as is clearly demonstrated in this study.

  1. Conclusion  (page 9, lines 300 to 302)

 “Reducing informal employment should be a key objective if public health institutions wish to improve the mental health of workers in Iberoamerica.”    Most countries have labor laws that protect workers’ rights and govern employment conditions. Reducing informal employment in this study would require a change in labor laws in some or all of the 13 Iberoamerican countries in the study area. This is not a possible undertaking and is not controlled by the variables used in this study.

  1. Public health programs are mostly government-funded (often underfunded) and prevention and control measures should be tailored to maximize the efficiency and effectiveness of interventions through the policy-backed strategic approach of targeting top public health priorities.  With regard prevention of a multidimensional problem like poor mental health, an integrated approach incorporating public health measures like prevention of substance abuse, prevention of gender-based discrimination, measures to improve occupational health, providing access to mental health services, prevention of adverse childhood experiences, etc. is more practical than attempt to change good governance and labor laws of more than a dozen countries across the globe.

References

  1. Allen, J., Balfour, R., Bell, R., & Marmot, M. (2014). Social determinants of mental health. International review of psychiatry, 26(4), 392-407.
  2. Alegría, M., NeMoyer, A., Falgàs Bagué, I., Wang, Y., & Alvarez, K. (2018). Social determinants of mental health: where we are and where we need to go. Current psychiatry reports, 20(11), 1-13.

Author Response

Reviewer 2

This reviewer commends the authors for their study titled, “Informal employment and poor mental health in a sample of 180,260 workers from 13 Iberoamerican countries. However, the following comments and questions need to be addressed.

Response: We thank the reviewer for the comments and suggestions. We have revised it and made appropriate changes in response to the comments and suggestions.

COMMENT #6.  Method (page 3, lines 84-93)

Design This study erroneously attempts to attribute population mental health to a single variable of informal employment.  According to the World Health Organization, “[c]ertain population subgroups are at higher risk of mental disorders because of greater exposure and vulnerability to unfavorable social, economic, and environmental circumstances, interrelated with gender.”1

Response: We completely agree that mental health has a multi-causal origin, resulting from exposure to poor social, economic and environmental conditions, but also pay work is an important determinant of health. In adition, it is well known that informal workers are a vulnerable group and that informal employment is not only related to poor working and poor employment conditions, but also to poor living, social, economic vulnerability and poor environmental conditions. Therefore, this study does not aim to attribute poor mental health solely to informal work, but rather to measure the strength of the association between informal employment and poor mental health in Iberoamerican countries adjusted for occupational variables and socio-demographic conditions. These results could modestly serve as a basis for new hypotheses and a baseline for future studies. To clarify this we have included new text (pag7 line 224):

“It is important to note that informal employment is related not only to poor working and employment conditions, but also to poor living conditions, social and economic vulnerability and poor environmental conditions. All these factors could be affecting the associa-tion. Previous studies have shown that national indicators of GDP, CO2 emissions, educational attainment, life expectancy, poverty rates, women´s labor market participation are strong-ly associate to informality rate [34]. ”

COMMENT #7.  Research Instruments “Mental health status was assessed by five different instruments. For the World Health Organisation- Five Well-Being Index (WHO-5), poor mental health was defined as a score under 13 [25]; for the 9-item Patient Health Questionnaire (PHQ-9) as a score of 10 or more [26]; for the psychological domain of the Abbreviated World Health Organisation  Quality of Life questionnaire (WHOQOL-Bref) as a score under 60 [27]; for the 12-item General Health Questionnaire (GHQ-12) following GHQ usual scoring method (0–0–1–1) as a score of 3 or more [28]; and for the affect domain of the Washington Group Extended 91 Set on Functioning (WG-ES) as a score of 4 in either depression or anxiety scales [29].”

As stated in the preceding paragraph mental health status was assessed using five different instruments across the 13 Iberoamerican countries and yet attempts to convince readers that the different measurements can be used to represent the same outcome variable of mental health.

Response: We agree that comparing the results obtained by different instruments can be an important limitation. However, all the instruments used in this study have been validated and are widely recognised and reliable for measuring mental health. Additionally, it is important to consider that in this study we compared the mental health of informal and formal workers within each country, i.e. using the same instrument for both groups of workers. To clarify this we have included new text limitation seccion (pag9 line 315):

“Another important limitation was that mental health status was assessed using five dif-ferent instruments. However, all instruments used have been validated, widely recognized and reliable measures of mental health. In addition, we measured the prevalence ratio of poor mental health between formal and informal workers within countries using the same instrument. Finally, given the data come from different countries and therefore from dif-ferent sources, the methodology might differ slightly. In addition, the surveys were con-ducted in different years, between 2012 and 2018. However, this study used data from each country to measure the difference between the mental health of informal and formal workers within countries. All these limitations should make us cautious when interpret-ing the results.”

COMMENT #8.  Operational Definitions (page 3, lines 94-99)

Regarding employment conditions, workers were classified according to social security affiliation into formal (affiliated) or informal (not affiliated). This was the case for Argentina, countries in Central America and Peru. When information on social security was not available, classification was based on affiliation to the health system: workers who are entitled to health services as holders were considered formal and those not entitled or entitled as beneficiaries to other family members were considered informal.”

As stated in the preceding paragraph this study does not use the same operational definition for the term temporary employment across the 13 Iberoamerican countries in the study area, and yet attempts to convince readers that the different definitions represent the same independent variable.

Response: We appreciate the reviewer's concern about the operationalisation of the main explanatory variable. The international Labour Organization (ILO) has defined informal jobs as those whose employment relationship is not subject to national labour legislation, income taxation, social protection or entitlement to employment benefits. However, the operativization of this variable is not always simple. The ILO in 2013 in an effort to standardise this published a statistical manual to measure informality focuses on technical issues to the production of statistics on informal employment and the informal sector. The same text shows the definition of informal jobs of employees in some countries. Although there are differences in the way they are defined and operationalised, they overlap with each other, and the results are close. Consequently, we have defined informal employment by the lack of a contract and/or lack of social security coverage (pension system and/or health care system). Although, these definitions are not exactley the same, they overlap. In many countries contract workers must be affiliated to a pension or health care system. Thus, not being affiliated is a close proxy to work without contract and therefore to informal employment. Furthermore, these are the best available data to measure informal employment and mental health in a national representative sample in each country. Anyway, we have included a new paragrah into limitation section in order to underline this discussion (pag 9 line 308):

“To operationalise informal employment, and according to the ILO aproach (ref), different criteria were used for each country, depending on the information available: affiliation to social security, to the health system or to having a contract. However, in many countries, workers with a contract must be affiliated to a pension or health system, then not being affiliated to the social security or health system is approaching to the same concept: informality.”

International Labour Organization (ILO). (2013). Measuring informality: a statistical manual on the informal sector and informal employment: Vol. April. http://www.ilo.org/wcmsp5/groups/public/---dgreports/---dcomm/---publ/documents/publication/wcms_222979.pdf

COMMENT #9.  The Studies Does Not Account for Other Major Mental Health Variables (page 3, lines 113 to 116

“Age was comprised into four categories (fewer than 24; 25-44; 45-64; and over 65 113 years old). Higher educational level achieved was categorized into less than primary 114 school; primary school; middle school; and higher education. Marital status was comprised into two categories: married or cohabiting; and single, separated or widowed.”

As stated in the preceding paragraph this study does not provide insight into child labor and its impact on adult mental health (notice the first age bracket is fewer than 24, not fewer than 18 years) and also ignores the enormous impact of adverse childhood experiences on adult mental health. Thus the study erroneously assumes poor mental health is the result of getting informal employment as adults. But it is important to note that other variables not accounted for in this study such as childhood adversity, discrimination (whether related to race/ethnicity, immigrant status, sexual orientation, and/or occupational status), nationality, and migration status have demonstrated significant negative impacts on mental health.2

Response: We agree that poor mental health is multidimensional problem and can be triggered by several variables including adverse childhood experiences and discrimination on adult mental health. Unfortunately, we do not have information to measure these variables. This study is a first approach that evidences a significant positive association and provides a baseline for future research. Additionally, as it is remarked in Methods section, the data used in this study are from workers over 15 years of age. This is because in most countries’ surveys do not collect data from minors due to ethical and methodological issues.

We clarify this point in the methods section

(pag 3 line 91)

“(…) This resulted in a sample of 180,260 workers aged 15 and over.”   

COMMENT #10.  Ecological Fallacy

Studies using geographical units involving several countries are inherently characterized by analytical difficulties and can often suffer from an "ecological fallacy," which attributes collective characteristics to very dissimilar individuals or groups or the lack of agreement on the power and methods of data analysis as is clearly demonstrated in this study.

Response: Despite this study use data from several countries, we think that it is not an ecological study. The unit of analysis is not the country but the individual. In each of the countries we use disaggregated data that allow us to calculate the probability, with a certain level of error, that an individual in informal employment reports poor mental health, adjusted for work-related and demographic variables.

COMMENT #11.  Conclusion  (page 9, lines 300 to 302)

 “Reducing informal employment should be a key objective if public health institutions wish to improve the mental health of workers in Iberoamerica.”

Most countries have labor laws that protect workers’ rights and govern employment conditions. Reducing informal employment in this study would require a change in labor laws in some or all of the 13 Iberoamerican countries in the study area. This is not a possible undertaking and is not controlled by the variables used in this study.

Response: We agree that moving from informal to formal employment as mentioned by several global institutions in their reports is not a simple step. However, we believe that it is an important and facible step in long term to improve working and employment conditions, which in turn would improve the health of workers. Furthermore, it is true that we cannot affirm this conclusion on the basis of our results. Therefore, we have modified the sentence as follows (pag9 line 316-318):

“Addressing informal employment could contribute to improving workers' mental health.”

COMMENT #12.  Public health programs are mostly government-funded (often underfunded) and prevention and control measures should be tailored to maximize the efficiency and effectiveness of interventions through the policy-backed strategic approach of targeting top public health priorities. With regard prevention of a multidimensional problem like poor mental health, an integrated approach incorporating public health measures like prevention of substance abuse, prevention of gender-based discrimination, measures to improve occupational health, providing access to mental health services, prevention of adverse childhood experiences, etc. is more practical than attempt to change good governance and labor laws of more than a dozen countries across the globe.

Response: We complitely agree with the reviewer that mental health is a multidimensional global problem that involves several social, economic, occupational and health policies areas. It must therefore be tackled on several fronts. In our opinion, work, as a key determinant of health, should be considered as one of the main areas of intervention, but not the only one, and always looking at the problem from a holistic approach.

To clarify this we have included new text.

“Work is a key determinant of workers' mental health, but other determinats such as social and living conditions should be included in the prevention policies.”

Round 2

Reviewer 1 Report

I really appreciate the authors for their revision works after I carefully read the responses to comments and compare the differences between the original and revised manuscript. The current version of manuscript has been much improved, particularly the study scope and included countries. The explanations for these issues are reasonable and acceptable. In my opinion, a study discovers some existing phenomena has already its value. The follow-up strategies or methods could be formulated with reference to the results of this study because different countries or regions have different situations. I think that the revised statement at the end of Abstract is more conservative and is also in line with reality.

Author Response

We greatly appreciate the reviewer’s positive comment